# Conformal Risk-Controlled Routing for Large Language Model

## Abstract

Recent advances in small-scale large language models have shown that compact models can successfully handle an expanding range of natural language and reasoning tasks. This progress opens the door to more affordable AI inference services by enabling broader use of cost-efficient models. However, existing approaches often fail to fully exploit small models due to fuzzy boundaries of their capabilities. In this paper, we propose a risk-controlled routing framework that dynamically selects among models of different scales, with a strong emphasis on maximizing the utility of smaller models. Our framework integrates supervised contrastive learning to enhance the separability of smaller-model capabilities and grounds its routing mechanism in conformal risk control, providing theoretical guarantees on system-level routing risk. Across benchmarks, our method delivers cost–accuracy performance that is comparable to or better than strong baselines, with an absolute accuracy improvement of $\sim 3.49\%$ at equal cost and up to $\sim 36\%$ cost reduction at comparable accuracy.

## 1 Introduction

Large language models (LLMs)(OpenAI, 2025a; DeepSeek-AI et al., 2025; Grattafiori et al., 2024) have progressed rapidly, demonstrating strong performance across a wide range of natural language and reasoning tasks. To increase accessibility, model families such as GPT(OpenAI, 2025b), Gemma(Team et al., 2025), and Qwen(Yang et al., 2025) are released in multiple scales, each with distinct accuracy-efficiency trade-offs. This diversity creates an opportunity to improve system-level efficiency: rather than relying exclusively on a single large model, queries can be adaptively routed to models of different scales. Realizing this potential requires solving a central systems problem: *LLM routing*(Ding et al., 2022; 2024; Hu et al., 2024). The goal is to design a mechanism that dynamically selects the most suitable model for each query, where suitability entails two criteria: achieving sufficient accuracy to solve the task and maintaining an inference cost affordable to most users.

Recent research on LLM routing mainly falls into two categories. The first is learning-based approaches, such as RouteLLM (Ong et al., 2025), HybridLLM (Ding et al., 2024), TO-router (Stripelis et al., 2024), BEST-route (Ding et al., 2025), and RouterDC (Chen et al., 2024b). The second is similarity-based approaches, where queries are embedded and routed based on their proximity or consistency in representation space, including clustering or nearest-neighbor retrieval (e.g., k-means–based partitioning in RouterBench (Hu et al., 2024)) and output-consistency methods such as Smoothie (Guha et al., 2024). These approaches do not require supervised training of a router but instead leverage the structural similarity among queries or the agreement among model outputs.

The primary limitation of current routing paradigms is their failure to fully exploit small, cost-efficient models, which are frequently bypassed even when capable. This under-utilization arises not from explicit design choices but from the inherent difficulty of predicting their performance. At its core lies a representation challenge: a small model's ability to correctly answer a query does not consistently align with its semantic representation. For example, two semantically similar queries may be mapped to nearby points in a standard embedding space, yet a small model may succeed on one and fail on the other (see Figure 1). The generic embeddings of prior work are insensitive to fine-grained differences in model capabilities, causing routers to be overly cautious and default to larger, more expensive models. Complementing this representational flaw is the risk-aware decision-

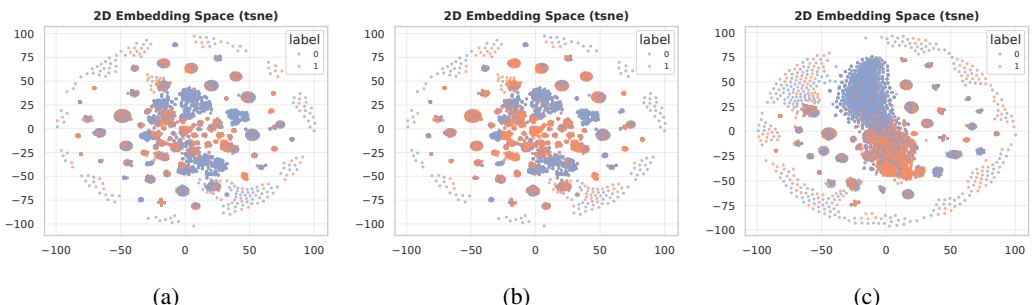

Figure 1: Embedding space separability (t-SNE). (a) Off-the-shelf embeddings: small-model correctness labels are heavily mixed. (b) Larger model: better but imperfect separation. (c) After SCL on the small model: clear delineation of answerable vs. unanswerable queries. Colors denote correctness (1/0).

making challenge. Assigning a complex query to an underpowered model wastes computation and may yield unexpected or incorrect responses, whereas sending a simple query to an expensive model incurs unnecessary cost. Existing methods typically rely on heuristic thresholds or fixed rules, but they lack a principled framework to formally quantify and control routing risk, leaving system-level behavior fragile and hard to guarantee.

To address these challenges, we propose *Conformal Risk-Controlled Routing* ($CR^2$), a framework that integrates capability-aware representation learning with principled risk control and cost-aware selection. Inspired by greedy algorithms, the first stage focuses on exploiting the utility of the smallest, most economical model. To tackle the core representation challenge, we employ supervised contrastive learning (SCL)(Khosla et al., 2020) to construct embeddings augmented with model-specific answerability, enabling the router to separate queries that are semantically similar but have different outcomes on the small model. Queries that cannot be confidently assigned to the small model are escalated to a second-stage router, which selects a candidate set of stronger models. To address the risk-aware decision-making challenge, we ground the routing process in the *Conformal Risk Control* (CRC). Specifically, we define a system-level risk function using candidate-set model-level false-positive rate (FPR) and calibrate a global candidate threshold under a held-out calibration set, providing formal guarantees for routing risk. Within the resulting candidate sets, a simple cost-aware rule selects the lowest-cost model, yielding a predictable and tunable accuracy–cost trade-off.

The main contributions of this work are summarized as follows:

- We propose $CR^2$, a two-stage routing framework that prioritizes the cost-efficient models. By leveraging supervised contrastive learning to refine embeddings, the router distinguishes semantically similar queries with divergent answerability, overcoming a key limitation of prior embedding-based methods.

- To the best of our knowledge, this is the first work to introduce CRC into LLM routing. By defining a bounded, composite routing loss and calibrating a global candidate threshold, $CR^2$ provides formal guarantees that the expected risk is provably bounded below specified level $\alpha$, while also yielding improved performance.

- Extensive experiments demonstrate that $CR^2$ establishes a new state of the art in LLM routing. It achieves an absolute accuracy improvement of approximately 3.49% (6% relative) over strong baselines such as EmbedLLM and single largest model, while simultaneously reducing overall operational cost.

## 2 RELATED WORK

### 2.1 MODEL ROUTING IN LLMS

Dynamic routing for efficiency spans multiple granularities: token-level mixtures-of-experts within a single forward pass (Fedus et al., 2022; Zhou et al., 2022; Li et al., 2025) and window-level

schemes such as speculative decoding (Leviathan et al., 2023; Lu et al., 2023; Chen et al., 2024c; Li et al., 2024). This work focuses on query-level routing, where an entire request is dispatched to one model from a pool. Existing methods include pre-generation routers that train lightweight selectors to pick a single model before inference (Ong et al., 2025; Ding et al., 2024; Feng et al., 2025; Stripelis et al., 2024; Ding et al., 2025) and post-generation cascades that escalate from cheaper to more expensive models until a quality criterion is met (Chen et al., 2024a). While effective, these approaches typically rely on fixed thresholds or heuristics and provide no distribution-free guarantees. Our framework complements this line by combining hierarchical routing with conformal risk control.

## 2.2 Capability-Aware Representations

Routing often hinges on representations that anticipate which model can solve a query. Early approaches embed models via accuracy profiles or simple classifiers to separate "easy" from "hard" queries (Zhuang et al., 2025; Ding et al., 2024). More recent work leverages contrastive objectives, either by jointly embedding queries and models (Chen et al., 2024b) or by modeling query–LLM relationships through transformer-based backbones (Jin et al., 2025). Other methods enrich embeddings with auxiliary signals, such as capability instructions that combine past performance and user prompts (Zhang et al., 2025b), or document-level context to capture knowledge shifts (Zhang et al., 2025a). However, these embeddings can conflate semantic similarity with *answerability*, leading to under-utilization of smaller, cost-efficient models. Our approach instead applies supervised contrastive learning to shape embeddings so that proximity reflects model-specific answerability, improving small-model utilization without sacrificing accuracy.

## 2.3 Risk-Aware Decision Making and Conformal Prediction

Beyond representation, routing is also a risk management problem. Conformal prediction provides distribution-free reliability guarantees, but classical coverage does not directly address cost–accuracy trade-offs. Conformal Risk Control (CRC) extends these tools to general bounded risks with finite-sample guarantees (Angelopoulos et al., 2024), and has been applied to mitigate hallucination in single-LLM settings (Overman et al., 2024; Chen et al., 2025). Other recent conformal methods include CP-Router, which uses uncertainty estimates for routing (Su et al., 2025), and another that optimizes risk and prediction set size during training (Noorani et al., 2024). To our knowledge, we are the first to introduce CRC into the routing pipeline itself: we calibrate a global candidate threshold so that the expected system-level routing risk—whose Stage-2 component is the candidate-set model-level false-positive rate—remains within a user-specified tolerance, while a cost-aware selector realizes efficiency gains.

## 3 Preliminaries

### 3.1 Problem Formulation

We study routing over a pool of large language models (LLMs) with heterogeneous sizes and inference costs. Let $\mathbb{Q}$ denote the space of queries and $\mathbb{M} = \{M_1, \ldots, M_K\}$ the available models. Each model $M_i$ is associated with an inference cost $c_i > 0$ and a correctness indicator $A_i(q) = \mathbf{1}[M_i(q) = y]$, where $M_i(q)$ is the output of $M_i$ on query $q$ and $y$ is the ground-truth answer; hence $A_i(q) \in \{0, 1\}$ indicates whether $M_i$ answers $q$ correctly. A routing strategy is a mapping $R : \mathbb{Q} \to \mathbb{M}$ that assigns a model $M_{R(q)}$ to each query $q$. The system-level correctness on $q$ is $A_{R(q)}(q)$. We evaluate a routing strategy $R$ by its expected accuracy

$$\text{Acc}(R) = \mathbb{E}_{q \sim \mathbb{Q}}\big[A_{R(q)}(q)\big], \tag{1}$$

and its expected cost

$$\text{Cost}(R) = \mathbb{E}_{q \sim \mathbb{Q}}\big[c_{R(q)}\big]. \tag{2}$$

The routing problem is thus a multi-objective optimization that maximizes accuracy while minimizing cost:

$$\max_R \big(\text{Acc}(R), -\text{Cost}(R)\big), \tag{3}$$

equivalently $\min_R \big(1 - \text{Acc}(R), \text{Cost}(R)\big)$, which induces a Pareto frontier.

**Remark.** In Section §4 we instantiate $R$ via a two-stage architecture that prioritizes the smallest model when safe and escalates otherwise; here we only establish notation and objectives.

## 3.2 Conformal Risk Control

CRC (Angelopoulos et al., 2024) is a statistical framework that generalizes classical conformal prediction from coverage guarantees to controlling the expectation of a general loss function. Given a base predictor, a calibration set $\{(X_i, Y_i)\}_{i=1}^n$, and a user-specified risk level $\alpha \in (0, 1)$, CRC provides a recipe for calibrating a parameter $\hat{\lambda}$ to ensure that the expected loss on a new test point does not exceed $\alpha$.

The framework operates on a family of predictors $C_\lambda(x)$ indexed by a parameter $\lambda \in \Lambda$. This parameter $\lambda$ controls the **conservativeness** of the predictor's output. We define a loss for each calibration example as

$$L_i(\lambda) = \ell(C_\lambda(X_i), Y_i). \tag{4}$$

A critical requirement of the framework is that the loss function $\ell$ must be **monotone non-increasing** with respect to $\lambda$. This ensures that a more conservative choice of $\lambda$ will not result in a higher loss. This property holds for many useful applications, such as controlling the false negative rate in multilabel classification or token-level F1 loss in question answering (Angelopoulos et al., 2024).

The goal of CRC is to select a data-driven threshold $\hat{\lambda}$ such that the following expected risk guarantee holds for a new test point $(X_{n+1}, Y_{n+1})$:

$$\mathbb{E}\left[L_{n+1}(\hat{\lambda})\right] \le \alpha. \tag{5}$$

CRC achieves this by calculating the empirical risk $\widehat{\mathcal{R}}(\lambda) = \frac{1}{n} \sum_i L_i(\lambda)$ on the calibration set and finding the least conservative $\lambda$ that satisfies a high-probability risk bound. For a loss bounded by $B$, this is typically:

$$\hat{\lambda} = \inf\left\{ \lambda \in \Lambda \;\middle|\; \frac{n}{n+1}\widehat{\mathcal{R}}(\lambda) \;+\; \frac{B}{n+1} \;\le\; \alpha \right\}. \tag{6}$$

This guarantee is distribution-free and holds for finite samples. When $\ell$ is chosen as the miscoverage loss, $\ell(C_\lambda(X), Y) = \mathbf{1}\{Y \notin C_\lambda(X)\}$, CRC reduces exactly to classical conformal prediction.

## 4 Methodology

In this section, we introduce Conformal Risk-Controlled Routing, a framework designed to address the dual challenges of representation and risk-aware decision-making in LLM routing. The core of our approach is to decompose the global routing task into two specialized sub-problems: (1) a high-precision binary prediction for the single, most cost-effective model, and (2) a multi-label prediction to identify capable models from the remaining expert pool. Crucially, instead of combining these stages with fragile heuristics, we unify them under the CRC framework. This is achieved by designing a global risk function and a corresponding decision algorithm, which together provide a provable guarantee that the system-level trade-off between cost and accuracy is explicitly controlled.

### 4.1 Problem Decomposition

We parameterize a hierarchical router by thresholds $\theta = (t_1, t_2)$:

$$R_\theta(q) = \begin{cases} M_1, & \text{if } s_1(q) \ge t_1, \\ \text{Select}\big(\mathcal{C}_{t_2}(q)\big), & \text{otherwise,} \end{cases} \tag{7}$$

where $s_1(q) \in [0, 1]$ estimates the success probability of $M_1$ on $q$, $\mathcal{C}_{t_2}(q) = \{ i \ge 2 : \hat{p}_i(q) \ge t_2 \}$ is a candidate set among larger models, and $\text{Select}(\cdot)$ is a cost-aware rule. Section 4.2 describes how to obtain $s_1(q)$; Section 4.3 details $\{\hat{p}_i(q)\}_{i \ge 2}$ and the CRC calibration of $\theta$.

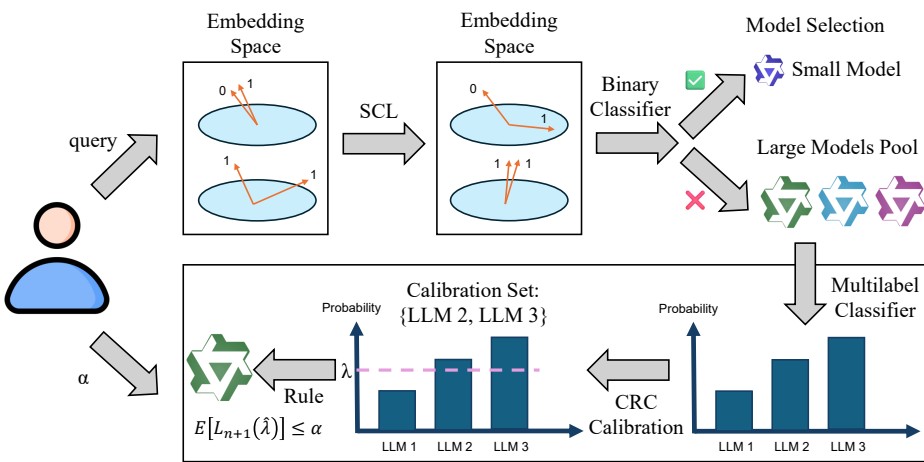

Figure 2: Overview of the $\mathrm{CR}^2$ framework. A binary classifier, operating on a capability-aware embedding space shaped by SCL, first attempts to route a query to the small model. If the query is rejected, a multilabel classifier scores the expert model pool. Finally, CRC uses these scores to calibrate a decision threshold that guarantees the final cost-optimal selection adheres to a user-specified risk tolerance.

## 4.2 Capability-Aware Filtering

The first stage of $\mathrm{CR}^2$ constructs a high-precision filter for the smallest model $M_1$, which predicts the binary correctness label $A_1(q) \in \{0,1\}$ for a given query $q$. This is achieved through a two-phase procedure: (i) fine-tuning a text encoder to learn capability-aware representations; and (ii) training a classification head on these embeddings.

**Architecture.** The module consists of a pretrained encoder $g_\theta$, a projection head $u_\varphi$, and a classification head $h_\psi$. To enrich the embedding for contrastive learning, $u_\varphi$ is designed as an *attention pooling projector*, which uses learnable query vectors to attend to token-level encoder outputs and yield more informative representations than mean pooling. The classification head is a two-layer MLP.

**Phase 1: Representation Learning via SCL.** We use SCL (Khosla et al., 2020) to endow the Stage-1 filter with a representation whose geometry reflects the *answerability* of the smallest model $M_1$, rather than mere semantic similarity.

**Setup.** Let $g_\theta$ be a pretrained text encoder and $u_\varphi$ a projection head (we use an attention-pooling projector). Given a query $q$, we form a normalized embedding

$$\boldsymbol{z}_q = \frac{u_\varphi\big(g_\theta(q)\big)}{\big\|u_\varphi\big(g_\theta(q)\big)\big\|_2}. \tag{8}$$

For a minibatch $\{(q_i, y_i)\}_{i=1}^{B}$, labels are $y_i = A_1(q_i) \in \{0,1\}$, where $A_1(q)$ indicates whether $M_1$ answers $q$ correctly (cf. Preliminaries).

**Supervised contrastive loss.** With temperature $\tau > 0$, the per-anchor SCL loss is

$$\mathcal{L}_i^{\mathrm{SCL}} = -\frac{1}{|\mathbb{P}(i)|} \sum_{p \in \mathbb{P}(i)} \log \frac{\exp\big(\boldsymbol{z}_i^\top \boldsymbol{z}_p / \tau\big)}{\sum_{a \in \mathbb{A}(i)} \exp\big(\boldsymbol{z}_i^\top \boldsymbol{z}_a / \tau\big)}, \tag{9}$$

where $\mathbb{P}(i) = \{\, p \neq i : \ y_p = y_i \,\}$ is the set of positives for anchor $i$ and $\mathbb{A}(i) = \{\, a \neq i \,\}$ the set of all non-anchor samples in the batch (anchors with $|P(i)| = 0$ are skipped). Minimizing $\mathcal{L}^{\mathrm{SCL}} =$

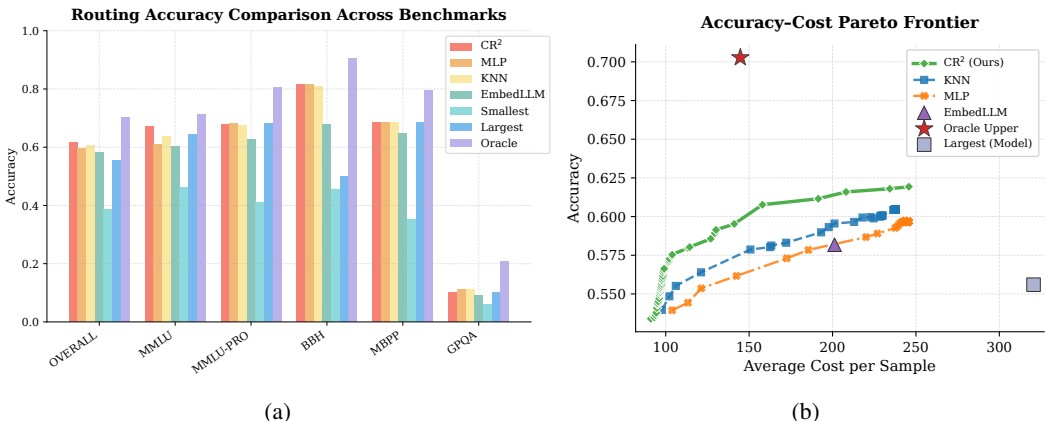

(a)                                                                                          (b)

Figure 3: (a) Routing accuracy of $CR^2$ compared to baselines. $CR^2$ router performs better almost across the whole test set. (b) Accuracy–cost trade-off of different routing strategies, where $CR^2$ achieves superior Pareto efficiency compared to baselines.

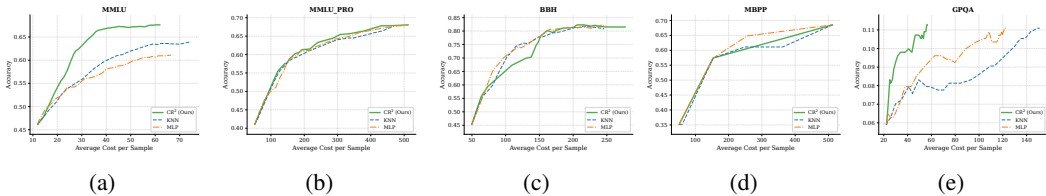

(a)                    (b)                    (c)                    (d)                    (e)

Figure 4: Accuracy–cost trade-off of different routing strategies per benchmark.

$\sum_i \mathcal{L}_i^{\text{SCL}}$ pulls together queries that $M_1$ handles similarly (both solvable or both unsolvable) and pushes apart those with different outcomes, thereby reshaping the embedding space to be capability-aware with respect to $M_1$.

**Classifier training on capability-aware embeddings.** After SCL fine-tuning, we *freeze* the encoder $g_\theta$ and projector $u_\varphi$, and train a lightweight classification head $h_\psi$ on the capability-aware representations. The head is trained to predict the success of model $M_1$, i.e., the binary label $y = A_1(q) \in \{0, 1\}$. It outputs two logits, and we convert their difference into a probability estimate via the sigmoid function:

$$s_1(q) = \sigma\big(\ell_1(q) - \ell_0(q)\big) \approx \Pr\big[A_1(q) = 1\big]. \tag{10}$$

Since the smallest model correctly handles only a limited proportion of queries, the training data for its success predictor suffers from a natural class imbalance. To mitigate this, the head is optimized by minimizing a class-weighted binary cross-entropy loss, where weights are determined by the inverse frequency of each class. At inference, the Stage-1 router routes to $M_1$ when $s_1(q) \geq t_1$ and otherwise escalates. We treat $t_1$ as a fixed gate (set on a held-out set) and use CRC (§4.3) to calibrate the Stage-2 candidate threshold so that the overall system-level routing risk satisfies the specified budget.

## 4.3 MULTILABEL CLASSIFICATION AND CRC CALIBRATION

For queries deferred by Stage 1 (i.e., $s_1(q) < t_1$), a multilabel head scores the remaining models

$$\hat{\mathbf{p}}(q) = \big(\hat{p}_2(q), \ldots, \hat{p}_K(q)\big) \in [0, 1]^{K-1}, \tag{11}$$

where $\hat{p}_i(q)$ estimates the probability that $M_i$ answers $q$ correctly. Let $y_{ij} = A_i(q_j) \in \{0, 1\}$ denote the ground-truth outcome for model $M_i$ on query $q_j$. Given a global threshold $\lambda \in [0, 1]$, we define the candidate set

$$\mathcal{C}_\lambda(q) = \{i \in \{2, \ldots, K\} : \hat{p}_i(q) \geq \lambda\}. \tag{12}$$

Our goal is to select $\lambda$ with a distribution-free, finite-sample risk guarantee *for the entire routed system* under a fixed gate $t_1$.

**Per-query loss and monotonicity.** On a held-out calibration set $\mathcal{D}_{\text{cal}} = \{(q_j, \mathbf{y}_j)\}_{j=1}^n$ (assumed exchangeable with test data), we define a bounded per-query loss

$$L_j(\lambda) = \begin{cases} 1 - y_{1j}, & \text{if } s_1(q_j) \geq t_1, \\ \dfrac{|\{\, i \in \mathcal{C}_\lambda(q_j) : \ y_{ij} = 0 \,\}|}{\max\big(1, \ |\{\, i \geq 2 : \ y_{ij} = 0 \,\}|\big)}, & \text{if } s_1(q_j) < t_1, \end{cases} \tag{13}$$

i.e., a misclassification indicator when routed to $M_1$, and the *model-level false-positive rate* within the candidate set otherwise. By construction $L_j(\lambda) \in [0, 1]$ and, holding $t_1$ fixed, $L_j(\lambda)$ is *non-increasing* in $\lambda$ (larger $\lambda$ shrinks $\mathcal{C}_\lambda$ and cannot add false positives). Hence the empirical risk

$$\widehat{\mathcal{R}}(\lambda) = \frac{1}{n} \sum_{j=1}^n L_j(\lambda) \tag{14}$$

is also non-increasing in $\lambda$. **We provide a formal proof that our composite loss function in Eq. equation 13 satisfies the crucial monotonicity property required by CRC in Appendix B.**

**Calibrating $\lambda$ via conformal risk control.** We apply CRC for bounded losses (here $B = 1$). For a user-specified tolerance $\alpha \in [0, 1]$, CRC selects

$$\lambda^* = \inf \left\{ \lambda \in [0, 1] : \underbrace{\frac{n}{n+1} \widehat{\mathcal{R}}(\lambda) + \frac{1}{n+1}}_{\text{CRC upper bound on } \mathbb{E}[L(\lambda)]} \leq \alpha \right\}. \tag{15}$$

Choosing the *smallest* feasible $\lambda^*$ maximizes candidate-set size under the same risk budget, preserving downstream cost opportunities while maintaining the distribution-free, finite-sample guarantee $\Pr\big(\mathbb{E}[L(\lambda^*)] \leq \alpha\big) \geq 1 - \delta$.

**Final selection rule and fallback.** At test time we use the fixed gate $t_1$ and set $t_2 = \lambda^*$. For $s_1(q) \geq t_1$, route to $M_1$; otherwise select

$$\text{Select}\big(\mathcal{C}_{t_2}(q)\big) = \arg\min_{i \in \mathcal{C}_{t_2}(q)} c_i \quad \text{(ties broken by larger } \hat{p}_i(q)\text{)}. \tag{16}$$

If $\mathcal{C}_{t_2}(q) = \varnothing$, we fall back to $\arg\max_{i \geq 2} \hat{p}_i(q)$ or a pre-specified robust model (see ablations). This policy, together with equation 15, yields distribution-free, finite-sample control of the expected composite risk in equation 13.

## 5 Experiment Results

### 5.1 Experimental Setup

We conduct a comprehensive set of experiments to evaluate the performance of our proposed method.

**Model Pool and Costs.** Our experiments utilize a pool of widely-used, open-source LLMs from Qwen3 family (Yang et al., 2025), which provides a realistic spectrum of capabilities and inference costs. Our model pool $\mathbb{M} = \{\text{Qwen3-1.7B}, \text{Qwen3-4B}, \text{Qwen3-8B}, \text{Qwen3-14B}\}$. We define the inference cost for each model based on the total number of input tokens according to official API price, normalizing them relative to the largest model Qwen3-14B. The relative costs are 0.15, 0.3, 0.5, 1.0 for Qwen3-1.7B, Qwen3-4B, Qwen3-8B, Qwen3-14B, respectively.

**Datasets.** Following the reproducible protocol of EmbedLLM (Zhuang et al., 2025), we evaluate on a diverse query corpus spanning six challenging benchmarks covering expert knowledge, multi-step reasoning, and coding: MMLU (Hendrycks et al., 2021), MMLU-Pro (Wang et al., 2024), GSM8K (Cobbe et al., 2021), Big-Bench Hard (BBH) (Suzgun et al., 2022), GPQA (Rein et al., 2023), and MBPP (Austin et al., 2021). To generate the ground-truth data, for each query $q$ from these benchmarks and each model $M_i \in \mathbb{M}$, we run inference using the lm-evaluation-harness (Gao et al., 2024).

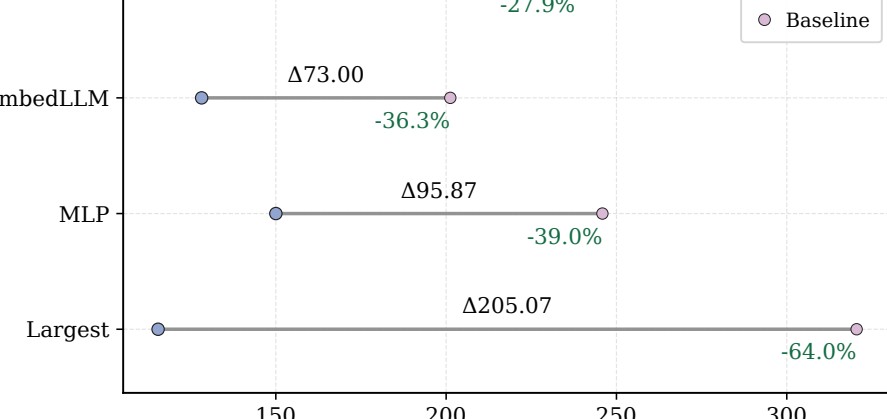

Figure 5: Cost dumbbell comparison at equal accuracy. $CR^2$ consistently achieves lower inference cost than baselines, with relative savings exceeding 60%.

**Baselines.** We compare our method against a comprehensive set of baselines to rigorously evaluate its performance:

- **EmbedLLM** (Zhuang et al., 2025): The current state-of-the-art learning-based router, which uses general-purpose embeddings to predict model performance.

- **Always-Smallest**: A simple heuristic that always routes to the cheapest capable model.

- **Always-Largest**: A heuristic that always routes to the most largest model.

- **Oracle**: A theoretical upper bound that assumes perfect knowledge of each model's answerability for every query. It always selects the cheapest model that is known to answer the query correctly, defining the Pareto frontier.

- **MLP**: A non-hierarchical baseline where a MLP is trained on top of general-purpose sentence embeddings. It acts as a multi-class classifier to select a single model from the pool based on the highest output score.

- **KNN** (Zhuang et al., 2025): A non-parametric baseline that performs nearest-neighbor voting over query–model correctness outcomes. Each model is implicitly represented by its historical correctness tuples, and for a new query, the classifier predicts performance based on the majority vote of its nearest neighbors. We refer to this approach as KNN throughout the text.

**Evaluation Metrics and Implementation Details.** We evaluate all methods on two primary metrics: Routing Accuracy (%) and Average Per-Sample Token Cost (Avg. Cost). For cost, we normalize API prices to that of the most expensive model, compute each sample's token cost as the normalized price-per-token times its total input tokens, and then average over all samples. The ideal method should achieve high accuracy at a low cost. For our method, we use a pretrained `all-MiniLM-L6-v2`(Wang et al., 2020) as the base sentence encoder, which is then fine-tuned using the supervised contrastive loss.

CRC is configured to control the expected system-level risk, ensuring it remains below the user-specified tolerance $\alpha$. For the main SOTA comparison, we set this risk level to $\alpha = 0.08$, though a broader analysis with varying $\alpha$ is also presented.

For each benchmark, we generate labels on its official training set, and then partition this labeled data into 80%/10%/10% splits for training, validating, and testing our router, respectively. Further implementation details, including all hyperparameters, are provided in Appendix A.

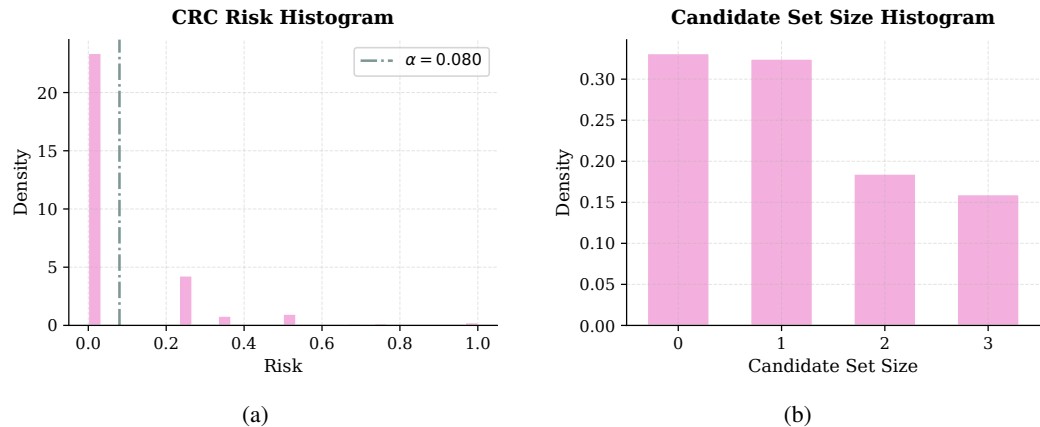

(a)                                                        (b)

Figure 6: Analysis of $CR^2$ routing. (a) Histogram of per-query risks under the calibrated CRC threshold, showing that the majority of samples lie well below the specified tolerance $\alpha$. (b) Histogram of candidate set sizes produced by the second-stage router, illustrating the distribution of model subsets considered at inference.

## 5.2 MAIN RESULTS

We now present the main experimental results, which show that $CR^2$ consistently outperforms strong baselines by achieving higher accuracy at equal cost and significantly reducing cost at equal accuracy.

First, in terms of routing accuracy, $CR^2$ achieves the best aggregate performance across five benchmarks (61.7%), with top or tied results on most individual tasks. As shown in Figure 3a, on BBH it surpasses the EmbedLLM baseline by 13.7 points, underscoring its strength on complex reasoning, while on MMLU it improves over the KNN baseline by 3.1 points, demonstrating robustness on knowledge-intensive evaluations. These results indicate that $CR^2$ narrows the gap to the oracle while preserving efficiency advantages over single-model deployments.

Second, when examining the accuracy–cost trade-off, $CR^2$ consistently defines the Pareto frontier. As shown in Figure 3b, its curve lies above all baselines and single-model settings, achieving higher accuracy at any given cost budget and substantially lower cost at a fixed accuracy. This highlights its ability to leverage smaller models effectively without sacrificing end-task performance.

Finally, we analyze the behavior of the risk calibration mechanism at inference time. Figure 6a shows that the calibrated router tightly controls the system's risk: per-query values are concentrated near zero, well below the user-specified tolerance of $\alpha = 0.08$. Figure 6b further illustrates how efficiency arises: for more than 65% of inputs, the router confidently selects a single candidate model (or none at all), while adaptively expanding to 2–3 candidates only on harder queries. This adaptivity explains how $CR^2$ remains both efficient and reliable in practice.

## 5.3 ABLATION STUDIES

We toggle each component while holding others fixed and report routing accuracy and average per-sample token cost in Table 1. Enabling the two-stage design improves accuracy from **58.49%** to **60.74%** (+2.25 pts) and reduces cost from 223.10 to 202.18 ($\sim$9.4%). Adding SCL lifts accuracy from **56.11%** to **58.34%** (+2.23 pts) and lowers cost from 201.56 to 196.04 ($\sim$2.7%), indicating clearer separability of answerable vs. unanswerable queries for the small model. CRC calibration trims cost from 239.71 to 220.47 ($\sim$8.0%) with essentially unchanged accuracy (61.05% vs. 60.97%).

Overall, the components are complementary: two-stage routing yields the largest gains, SCL sharpens the Stage-1 decision boundary, and CRC delivers reliable cost reductions under a risk budget.

## 6 CONCLUSION

In this work, we introduce $CR^2$, a novel hierarchical routing framework that learns capability-aware representations via supervised contrastive learning and, in a first for this domain, utilizes CRC to provide provable guarantees on the cost-accuracy trade-off. Experiments demonstrate that $CR^2$ establishes a new state-of-the-art, significantly improving both accuracy and cost-efficiency over strong baselines. By making the deployment of diverse LLMs more reliable and economically viable, our work represents a concrete step toward the affordable AI.

## 7 ETHIC STATEMENT

Our routing system could exacerbate fairness and bias issues if queries about sensitive topics are sent to smaller models that have not received the same safety alignment as larger models. While our framework does not inherently introduce bias, fairness depends on the quality and tuning of the model pool. Future work should explore routing criteria that explicitly account for fairness and safety.

## 8 REPRODUCIBLITY STATEMENT

We provide sufficient information to facilitate the reproduction of our results. The core code will be included in the supplementary material. Detailed implementation specifics, including model architectures, training procedures and hyperparameters, are described in the Appendix A. The full code will be publicly released on GitHub upon acceptance of the paper.

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

## A    APPENDIX 1: TRAINING IMPLEMENTATION DETAILS

### A.1    STAGE-1: CAPABILITY-AWARE FILTERING

**SCL**    We initialize the text encoder with `sentence-transformers/all-MiniLM-L6-v2` and its tokenizer (max sequence length 512; EOS used as `pad_token` when absent). On top of the encoder we add an attention-pooling projector with two learned queries followed by a linear layer to a 384-d embedding; outputs are L2-normalized. Mini-batches contain 256 examples and are composed with a class-balanced sampler to stabilize SCL. We fine-tune the encoder and projector for 15 epochs using AdamW (LR $1 \times 10^{-4}$ on projector/encoder, weight decay 0.01), cosine annealing with warm restarts ($\eta_{\min} = 1 \times 10^{-6}$), mixed precision on GPU (bfloat16), and gradient-norm clipping on the projector (max-norm 5.0). All runs use seed 42 and Weights&Biases for logging.

**Binary classifier training on frozen embeddings.**    After SCL, both encoder and projector are frozen. We train a lightweight MLP head (LayerNorm $\rightarrow$ Linear($2d$) $\rightarrow$ GELU $\rightarrow$ Dropout(0.1) $\rightarrow$ Linear(2)) for 10 epochs with AdamW (LR $1 \times 10^{-3}$), the same cosine scheduler, and gradient clipping (max-norm 1.0). Class imbalance is handled via inverse-frequency class weights. This gate is fixed at deployment.

### A.2    STAGE-2: MULTILABEL CLASSIFICATION AND CRC CALIBRATION

**Multilabel Classifier Training.**    We train a multilabel classifier to score the remaining models $\{M_i\}_{i=2}^{K}$ using a frozen MiniLM encoder and a lightweight head. Concretely, we instantiate `sentence-transformers/all-MiniLM-L6-v2` and freeze all backbone parameters. Inputs are tokenized with the corresponding tokenizer (padding enabled; `max_len=512`; EOS used as `pad_token` if absent). The head is an MLP (LayerNorm $\rightarrow$ Dropout(0.1) $\rightarrow$ Linear($d$, $4d$) $\rightarrow$ GELU $\rightarrow$ Dropout(0.1) $\rightarrow$ Linear($4d$, $K-1$)), trained with `BCEWithLogitsLoss`. Batches contain 64 examples per GPU; we run distributed data-parallel training on 4 NVIDIA 4090 GPUs via `torchrun`, yielding an effective global batch size of 256. Optimization uses AdamW (LR $= 1 \times 10^{-3}$, weight decay $= 0.01$) for 20 epochs with a cosine schedule and 3% warmup. Gradients are clipped at 1.0.

To mitigate label imbalance across the $(K-1)$ binary targets, we use inverse-frequency weighted sampling, where per-example weights are the clipped ($[0.2, 5.0]$) sum of inverse per-class positive rates. Validation runs every epoch with distributed aggregation. Unless otherwise specified, we set the random seed to 42 and use 4 dataloader workers per process. The resulting probabilities serve as inputs to the CRC calibration step that determines the stage-2 candidate threshold used at deployment.

## B    APPENDIX 2: PROOF OF MONOTONICITY FOR THE COMPOSITE LOSS FUNCTION IN CRC

Here, we formally prove that the composite loss function $L_j(\lambda)$ defined in Equation equation 13 of the main text is monotone non-increasing with respect to the threshold $\lambda \in [0, 1]$. This property is a prerequisite for the application of the Conformal Risk Control framework.

**Proposition 1.** *The composite loss function $L_j(\lambda)$ is monotone non-increasing with respect to $\lambda$.*

*Proof.*    To prove that $L_j(\lambda)$ is monotone non-increasing, we must show that for any pair of thresholds $0 \leq \lambda_1 < \lambda_2 \leq 1$, it holds that $L_j(\lambda_2) \leq L_j(\lambda_1)$. The loss function is defined piece-wise based on the routing decision for a given query $q_j$, so we analyze each case.

**Case 1: The query is handled by Stage 1** ($s_1(q_j) \geq t_1$)**.** In this case, the loss is defined as $L_j(\lambda) = 1 - y_{1j}$. This value is a constant with respect to $\lambda$, as it does not depend on the threshold. A constant function is, by definition, monotone non-increasing. Thus, $L_j(\lambda_2) = L_j(\lambda_1)$, and the condition is satisfied.

**Case 2: The query is handled by Stage 2** $(s_1(q_j) < t_1)$**.** In this case, the loss is the model-level FPR:

$$L_j(\lambda) = \frac{|\{\, i \in \mathcal{C}_\lambda(q_j) : \; y_{ij} = 0 \,\}|}{\max\left(1, \; |\{\, i \geq 2 : \; y_{ij} = 0 \,\}|\right)}.$$

Let us analyze the components of this fraction. The denominator, $D = \max\left(1, |\{\, i \geq 2 : \; y_{ij} = 0 \,\}|\right)$, is a positive constant for a given query $q_j$, as it depends only on the ground-truth outcomes, not on $\lambda$.

The numerator, $N(\lambda) = |\{\, i \in \mathcal{C}_\lambda(q_j) : \; y_{ij} = 0 \,\}|$, is the number of incorrect models included in the candidate set. To prove that $L_j(\lambda)$ is non-increasing, it is sufficient to prove that the numerator $N(\lambda)$ is non-increasing.

The candidate set is defined as $\mathcal{C}_\lambda(q_j) = \{\, i \in \{2, \ldots, K\} : \; \hat{p}_i(q_j) \geq \lambda \,\}$. Consider our two thresholds such that $0 \leq \lambda_1 < \lambda_2 \leq 1$. For any model $i$ to be in the set $\mathcal{C}_{\lambda_2}(q_j)$, its score must satisfy $\hat{p}_i(q_j) \geq \lambda_2$. Because $\lambda_2 > \lambda_1$, this condition implies that $\hat{p}_i(q_j) > \lambda_1$, which in turn means that model $i$ must also be a member of the set $\mathcal{C}_{\lambda_1}(q_j)$.

Therefore, the candidate set at the higher threshold is a subset of the candidate set at the lower threshold:

$$\mathcal{C}_{\lambda_2}(q_j) \subseteq \mathcal{C}_{\lambda_1}(q_j).$$

The numerator $N(\lambda)$ counts the number of incorrect models within the candidate set. Let $I_{\text{incorrect}} = \{i \geq 2 : y_{ij} = 0\}$ be the set of all incorrect models for query $q_j$. The numerator can be written as $N(\lambda) = |\mathcal{C}_\lambda(q_j) \cap I_{\text{incorrect}}|$.

Since $\mathcal{C}_{\lambda_2}(q_j)$ is a subset of $\mathcal{C}_{\lambda_1}(q_j)$, the intersection of this smaller set with $I_{\text{incorrect}}$ must also be a subset of the intersection of the larger set with $I_{\text{incorrect}}$:

$$\mathcal{C}_{\lambda_2}(q_j) \cap I_{\text{incorrect}} \subseteq \mathcal{C}_{\lambda_1}(q_j) \cap I_{\text{incorrect}}.$$

The cardinality of a subset cannot be greater than the cardinality of the set that contains it. Thus, it follows that $N(\lambda_2) \leq N(\lambda_1)$. As the denominator is a positive constant, we have shown that the loss is non-increasing for this case as well.

**Conclusion.** Since the loss is monotone non-increasing in both cases, the composite loss function $L_j(\lambda)$ is proven to be monotone non-increasing with respect to $\lambda$ over its entire domain. $\square$

## C APPENDIX 3: ABLATION STUDIES

Table 1: Ablation study of Two Stage Routing, SCL and CRC.

|  | Accuracy (%) | Avg. Cost |
|---|---|---|
| w/o Two Stage Routing | 58.49 | 223.10 |
| w/ Two Stage Routing | 60.74 | 202.18 |
| w/o SCL | 56.11 | 201.56 |
| w/ SCL | 58.34 | 196.04 |
| w/o CRC | 61.05 | 239.71 |
| w/ CRC | 60.97 | 220.47 |

## D APPENDIX 4: THE USE OF LARGE LANGUAGE MODELS

We used OpenAI ChatGPT and Google Gemini (Deep Research) strictly as productivity aids. Concretely, they were used to (i) polish wording and improve stylistic clarity of drafts, and (ii) help scope the literature at the project outset by suggesting search terms and candidate papers. No parts of the methods or results were generated by LLMs. Every citation surfaced during scoping was manually verified against primary sources, and no model-generated references were accepted. No confidential data were shared with the tools. The authors take full responsibility for the content of this paper.

Table 2: Sensitivity of CRC to the calibration sample size ($\alpha = 0.1$). The risk remains close to $\alpha$ while accuracy shows only small variations.

| Calibration size $N_c$ | $\lambda$ | Acc | Avg token cost | Mean risk | Mean set size |
|---|---|---|---|---|---|
| 100 | 0.9053 | 0.6035 | **191.651** | 0.1090 | **1.4267** |
| 250 | 0.9286 | **0.6082** | 206.417 | **0.0907** | 1.2233 |
| 500 | 0.9199 | 0.6069 | 201.863 | 0.0979 | 1.3107 |
| 1000 | 0.9266 | 0.6074 | 202.217 | 0.0931 | 1.2523 |

Table 3: CRC performance under different $\alpha$ values (refit). The observed risk closely matches the target $\alpha$.

| Variant | $\alpha$ | $\lambda$ | Acc | Avg token cost | Mean risk | Mean set size |
|---|---|---|---|---|---|---|
| CRC@$\alpha$ | 0.05 | 0.9827 | **0.6095** | 233.213 | 0.0520 | 0.5640 |
| CRC@$\alpha$ | 0.10 | 0.9253 | 0.6074 | 202.176 | **0.0949** | 1.2781 |
| CRC@$\alpha$ | 0.15 | 0.8712 | 0.5952 | **150.947** | 0.1497 | **1.7532** |

# E APPENDIX 5: OUT-OF-DISTRIBUTION EVALUATION

Fig. 7 shows the accuracy–cost Pareto frontier evaluated on a combined out-of-distribution test set constructed from PIQA and ARC-Easy, which the router was not exposed to during training. Each point represents the average inference cost per sample under a specific routing configuration. The results reflect the model pool's performance and routing behavior in a distribution-shift setting rather than in-distribution generalization.

Our method $CR^2$ (green diamonds) forms a competitive Pareto frontier relative to KNN and MLP routers across the evaluated cost range. The Largest Model baseline (grey square) provides an upper-cost reference, while Oracle Upper Bound (red star) denotes the idealized maximum achievable accuracy if the best model were chosen per sample. The frontier of EmbedLLM is also plotted for comparison.

Overall, this OOD evaluation illustrates that routing continues to extract favorable accuracy–cost trade-offs even when the input distribution differs from training, and that the achievable frontier may exceed the standalone performance of the largest model.

**Accuracy-Cost Pareto Frontier**

Figure 7: Accuracy–cost trade-off of different routing strategie under OOD settings.

