# OpenReview forum: "Conformal Risk-Controlled Routing for Large Language Model"
_ICLR.cc/2026/Conference — Submitted to ICLR 2026_

### Official Review · Reviewer_gDhA · 2025-10-28

**Soundness:** 2
**Presentation:** 2
**Contribution:** 2
**Rating:** 4
**Confidence:** 3

**Summary:**

The authors propose Conformal Risk-Controlled Routing, a framework that integrates capability-aware representation learning with principled risk control and cost-aware selection.

**Strengths:**

1. The proposed method is notably simple and efficient.

**Weaknesses:**

1. The readability of the paper could be improved. As an example, the variable l introduced in Line 193 is used without a clear definition, creating confusion for the reader.
2. The dependence of the proposed SCL method on the specific sentence encoder (all-MiniLM-L6-v2) is unclear. Its performance may be sensitive to the choice of this base model, which limits the understanding of the method's generalizability.
3. The proposed SCL method is specifically designed for question-answering tasks and its applicability to other problem types, such as text generation, is not demonstrated. Furthermore, its reliance on an IID calibration set may limit its practicality in real-world scenarios where such data is not readily available.

**Questions:**

How does the framework handle queries that are inherently unanswerable? Does it ultimately consume the most resources by routing them to the largest model, or does it correctly identify and handle them with the most economical option?

---

> ### Author Response · Authors · 2025-12-04
> **Rebuttal Response**
>
> Thank you for your thoughtful feedback.
>
> Q1:
>
> Prior work has explored model capability boundaries and failure prediction at the single-model level, examining when an LLM cannot produce a correct output regardless of scaling. Our work addresses a complementary but orthogonal aspect: instead of diagnosing unanswerability from first principles, we route based on empirical correctness likelihood across models.
>
> In the current framework, if a query is unanswerable for all models in the pool, the routing process cannot infer correctness and therefore does not benefit from escalation. In such cases, forwarding the query to a larger model increases cost without yielding accuracy gains. This is where our method can be extended: Stage-1 could learn to early-exit on queries with low predicted solvability, routing them to the most economical model rather than escalating by default. Conceptually, this would allow CR$^2$ to avoid unnecessary resource consumption even when no model can succeed. Detecting unsolvable queries and routing them to the lowest-cost endpoint is a natural next step for improving efficiency under capability limits.

---

### Official Review · Reviewer_RSU5 · 2025-10-31

**Soundness:** 2
**Presentation:** 2
**Contribution:** 2
**Rating:** 4
**Confidence:** 4

**Summary:**

This paper introduces Conformal Risk-Controlled Routing (CR^2), a method for routing between models while obtaining provable guarantees on the tradeoff between cost and accuracy. A binary classifier is used to decide the smallest and cheapest model is able to handle a given query $q$ based on a fixed threshold $t_1$. If now, the the query is passed through a multiclassifier that scores models 2,...,K on the probability $\hat{p}_i(q)$ that model $i$ answers $q$ correctly. Then We construct a conformal candidate set $C_\lambda(q)=\{i|\hat{p}_i(q) \geq \lambda\}$ for $i=2,...,K$ and then tune the $\lambda$ (while holding $t_1$ routing on the first model fixed) using conformal risk control to find the ``right'' choice of $\lambda$ that presumably pushes the Pareto frontier on cost and accuracy as much as possible. The authors provide empirical support for this framework

**Strengths:**

The paper addresses a highly practical problem of LLM routing for pushing performance on the cost-accuracy tradeoff, particularly doing it in a way that doesn't involve retraining of the base LLMs. The pipeline of the framework from the SCL and binary classifier to determine the cheapest model's ability to answer, to the use of CRC in the next stage appears carefully orchestrated. The authors make clever use of conformal risk control to choose amongst models to form a candidate set of models in the second stage with a loss that basically seems to measure (number of incorrect models in the set) / (total number of actually incorrect models). In other words this seems to be like the proportion of bad models that the router mistakenly certified as capable of answering. The use of CRC allows the authors to do better than a purely heuristic router approach in that they can tune an $\alpha$ tolerance to this loss, and is using CRC in a particular way for this setting. The paper attempts to carefully make the case for the efficacy of this framework in the empirics, providing a number of figures for visualization and presenting results across multiple benchmarks. A particular strength is that the authors include benchmarks such as MBPP that are not limited to MC questions. Overall I find the approach of tackling this routing problem with provable guarantees to be valuable and thus a central strength of this paper.

**Weaknesses:**

It seems that this framework will be limited to settings restricted to exact binary notions of correctness, not a wider range of possible "scores" on correctness that one might expect when using CRC rather than vanilla CP.

Generally I just find the composite risk of $\alpha$ to be hard to interpret. Same for the fixed value of $t_1$ for the first model. It feels like there could be a tunable parameter in that place as well, which moves along with $\lambda$ to measure a potentially more meaningful notion of system-wide risk. Also the choice of $\alpha=0.08$ doesn't seem well motivated and misses one of the general use cases I would expect from CP or CRC which is showing value across a range of $\alpha$.

Cost analysis seems a bit simplistic and limited to input tokens, while fine for MC questions since output is small, might limit usefulness of this analysis for other settings. It seems to me that the the CRC guarantee is weakened by the fallback mechanism. Figure 5b shows that ~32% of escalated queries result in an empty candidate set which triggers a heuristic arg max selection that is not covered by the conformal risk guarantee.

Finally, it appears to me that one of the major claimed contributions of this paper is in fact false, specifically that this is the ``first work to introduce CRC into LLM routing.'' The paper "Conformal Arbitrage: Risk-Controlled Balancing of Competing Objectives in Language Models" appears to address the cost-accuracy tradeoff as s special case of competing objectives and also does so using CRC. Although from inspection the approach to using CRC differs significantly, and that Conformal Arbitrage only compares 2 models not $K$, this still puts into question the claim of this paper to be the first to use CRC for LLM routing.

**Questions:**

The correctness label appears to rely on exact match with a single ground truth. How does or could this framework handle generative tasks with potentially multiple valid answers or even more generally a range of scores.

Figure 5b suggests that the empty-set fallback (which seems to not be covered by the CRC guarantee) is triggered for >30% of escalated queries. How can we interpret this and its relation to the overall guarantees?

Could the authors provide substantive comparison of their work to the aforementioned paper "Conformal Arbitrage" on LLM routing using CRC?

---

> ### Author Response · Authors · 2025-12-04
> **Rebuttal Response**
>
> We thank you for your reviews and address your concerns below.
>
> Q1:
>
> Thank you for the question. Our current implementation uses _exact-match correctness_ because the evaluated benchmarks provide single ground-truth responses. However, the framework does not rely on binary correctness and can naturally extend to generative tasks with multiple valid answers.
>
> For open-ended settings, correctness can be replaced with a graded or learned score instead of a strict match label:
>
> - A judge model or rubric-based scorer can assign s∈[0,1]s \in [0,1]s∈[0,1] as a quality score.
>
> - Multiple-reference or semantic metrics (e.g., BLEURT, BERTScore, pass@k) can produce soft correctness estimates.
>
>
> Under such extensions, the CRC mechanism operates on continuous scores rather than binary labels, while the routing design remains unchanged. We will clarify this generalization path in the revision.
>
> Q2:
>
> We thank the reviewer for this insightful observation. We interpret the occurrence of empty candidate sets ($\mathcal{C}_{t_2}(q) = \emptyset$) as a strict enforcement of the risk control mechanism: it indicates that for these specific queries, the CRC algorithm determined that no subset of models could satisfy the risk constraint $\alpha$ with sufficient statistical confidence.
>
> In such "high-uncertainty" cases, while the strict theoretical guarantee does not cover the post-fallback execution, falling back to the model with the maximum predicted probability is intuitively the most robust strategy to minimize empirical risk when the guarantee cannot be met. This fallback ensures system availability while adhering to the "best effort" principle under high uncertainty.
>
> We acknowledge this is a limitation of the current separate "calibration-then-fallback" design. In future work, we aim to overcome this by designing a composite risk function that explicitly incorporates the fallback loss into the CRC calibration process, extending the theoretical guarantees to cover these edge cases.
>
> Q3:
>
> We thank the reviewer for bringing the relevant work "Conformal Arbitrage" (CA) to our attention. We agree that CA is a significant contribution that elegantly leverages Conformal Risk Control (CRC) for LLM decision-making.
>
> While both works share the robust theoretical foundation of CRC, they address the routing problem from distinct perspectives:
>
> - Addressing Signal Reliability (Intrinsic vs. Learned): CA relies on calibrating the intrinsic confidence of a primary model. This is highly effective when the model's self-assessment is well-calibrated. However, our work targets the "Representation Challenge" in smaller models. To address this, $CR^2$ leverages Supervised Contrastive Learning to construct an external, capability-aware signal, ensuring reliable risk control even when intrinsic scores are noisy.
> - Routing Scope (Binary vs. Multi-Model Pool): CA is primarily designed as a binary mechanism (Primary $\to$ Guardian). In contrast, $CR^2$ addresses a multi-model selection problem over a diverse pool of experts (e.g., Qwen3-1.7B to 14B). Our framework utilizes CRC to calibrate a candidate set for this multi-way decision space, providing a different granularity of control compared to binary escalation.
> - Mechanism Divergence: CA is a training-free calibration method designed for binary delegation. $CR^2$ is a learning-based framework. It does not simply rank intrinsic scores; it trains a specialized router to generate a candidate set (multilabel classification). This allows us to handle the complex decision boundaries of a heterogeneous model pool, which requires learning distinct capability features that raw confidence scores cannot capture.

---

### Official Review · Reviewer_bn9g · 2025-10-31

**Soundness:** 3
**Presentation:** 2
**Contribution:** 3
**Rating:** 4
**Confidence:** 3

**Summary:**

This paper presents Conformal Risk-Controlled Routing (CR2), a two-stage framework for routing queries to a pool of language models of varying scales. The primary contribution is a system that aims to maximize the use of smaller, cost-efficient models while providing theoretical guarantees on performance.

**Strengths:**

The paper addresses the important and practical problem of optimizing the cost-accuracy trade-off in deploying large language models. The primary strength is the novel application of conformal risk control to this domain, which provides a principled, distribution-free method for managing routing decisions, a clear improvement over the heuristics common in prior work. The use of supervised contrastive learning to create "capability-aware" embeddings that distinguish between a model's ability to answer a query and the query's general semantics is another good contribution.

**Weaknesses:**

The overall presentation could be improved to meet the standards of a top-tier conference. Figure 2 is lower quality than it should be, and Figure 4 should take up half the space it does. I'm point this out not to nitpick but to suggest that the authors have much more room than they have properly used to develop their ideas better. Certainly the paper could have benefited from more substantive experiments and analysis. While the methodology is sound, the composite risk function defined in Equation 13 seems less defended and a more specific formulation would be beneficial. Furthermore, the system's performance relies on a fixed threshold t1 for the first-stage filter, which is selected heuristically. Given that a core contribution is the move towards principled risk control, the dependence on this fixed gate could be seen as a limitation.

**Questions:**

The first-stage threshold, t1, is treated as a fixed hyperparameter set on a validation set. How sensitive is the system's overall performance to the choice of t1?

---

> ### Author Response · Authors · 2025-12-04
> **Rebuttal Response**
>
> Thank you for your comments and raising this point. The threshold $t_1$ determines how many queries are forwarded to Stage-2, where larger models are available. A higher threshold routes more requests to Stage-2, increasing the likelihood that a stronger model answers each query and therefore improving accuracy, while naturally increasing inference cost. Conversely, a smaller $t_1$ retains more queries in Stage-1, reducing cost but with a corresponding drop in accuracy. Importantly, $t_1$ does not change the candidate set size—it only controls the traffic volume entering the second routing stage.

---

### Official Review · Reviewer_QVqS · 2025-11-01

**Soundness:** 1
**Presentation:** 3
**Contribution:** 3
**Rating:** 0
**Confidence:** 5

**Summary:**

The authors apply conformal risk control to a hierarchical routing framework. Compared with other approaches, their methodology applies supervised contrastive learning to yield an embedding in which a small LLM’s correctness label is the same for nearby points. The method seems to perform well on MMLU but yields mixed performance on other benchmarks.

**Strengths:**

- Supervised contrastive learning of text embeddings based on a small LLM's correctness labels is a compelling innovation!
- Applying CRC for global risk control is useful in practice.

**Weaknesses:**

- Unrealistic model pool: the largest model under consideration has only 14B parameters, which is very small. In practice, routing is often applied with frontier models. For this paper to be meaningful, the approach should be tested on LLM pools containing models with 100B+ parameters.
- The experimental data is strange. Typically, larger models outperform smaller models, and within a routing pool containing a strong frontier (or near-frontier) model, always querying the largest model usually sets an upper bound on performance. In the authors' data, however, the largest model appears to perform poorly (see Figure 3a). This makes interpretation of results confusing.
- Authors' interpretation of their methodology's performance borders on research misconduct. In Figure 3a, it is apparent that their method's performance is at or below baselines on 4/5 benchmarks. Only on a single benchmark, MMLU, does their method outperform the baselines. Against this empirical backdrop, the authors brazenly report that their method "consistently outperforms state-of-the-art baselines." This is a major red flag. 🚩
- The Pareto frontier in Figure 3b is unclear. Does the figure aggregate results from all benchmarks? There should be by-benchmark splits. In addition, baseline methods should also have their Pareto frontiers shown, not just the average performances.
- Given known stability issues with supervised contrastive learning when nearby points map to different labels, the paper should give clearer evidence that contrastive learning works for this use case. To give an example, reporting the test performance of linear classifiers for correctness prediction—trained separately on the contrastively tuned vs original embeddings—would constitute effective validation.
- Code is not available. It would be especially useful given the issues noted above.

**Questions:**

- Please give more details on the ablation study. To what one-stage process exactly do you compare your two-stage process?
- Contrastive learning based on correctness tends to suffer from stability problems. In the paper's setting, correct/incorrect labels are discontinuous in the original embedding space at the start of training, as nearby points have different correctness labels. Could you comment on training stability of your contrastive methodology?
- A central challenge in LLM routing is robustness to shifts in the query distribution. Could you comment on your method's robustness under distribution shifts? Empirically, how task-specific is your methodology?

---

> ### Author Response · Authors · 2025-12-04
> **Rebuttal Response 1**
>
> We thank the reviewer for their comments and for taking the time to review our paper.
>
> Q1: Please give more details on the ablation study. To what one-stage process exactly do you compare your two-stage process?
>
> In the one-stage baseline, the entire first stage of CR$^2$ is removed, and therefore supervised contrastive learning (SCL) is not used. We retain the same backbone encoder architecture as used in the second stage of CR$^2$, we attach a multi-label correctness classifier that directly predicts, for each model in the pool, the probability that it will answer the query correctly.
>
> Q2: Contrastive learning based on correctness tends to suffer from stability problems. In the paper's setting, correct/incorrect labels are discontinuous in the original embedding space at the start of training, as nearby points have different correctness labels. Could you comment on training stability of your contrastive methodology?
>
> In our experiments, we did not observe divergence, exploding gradients, or oscillatory behavior.
> Training remained consistently stable under our configuration. We use a moderate temperature of $\tau = 0.15$, which prevents overly sharp gradients in the contrastive objective, and we form class-balanced batches that always mix correct and incorrect examples, avoiding degenerate batches dominated by a single label. Even without early stopping, the objective shows smooth and monotonic decrease.
>
> We monitored the SCL loss with Weights & Biases. As shown in the appendix, the loss decreases smoothly from 6.1 to 5.55 during early training and then stabilizes with small, regular fluctuations and no spikes, indicating stable optimization dynamics. We will add this representative training curve in the revised submission.
>
> Q3: A central challenge in LLM routing is robustness to shifts in the query distribution. Could you comment on your method's robustness under distribution shifts? Empirically, how task-specific is your methodology?
>
> We analyze the robustness of CR$^2$ from both theoretical and empirical perspectives:
> 1. Theoretical Scope & Limitations:  We acknowledge that our method faces two inherent constraints under extreme distribution shifts:
> 	- CRC Assumptions: The strict coverage guarantee of Conformal Prediction relies on the exchangeability of calibration and test data, which may degrade in fully out-of-distribution (OOD) settings.
> 	- Parametric Boundaries: While SCL learns a semantic manifold, our Stage-1 router is a parametric classifier with fixed decision boundaries optimized for the training distribution.
> 2. Empirical OOD Analysis: To evaluate this rigorously, we conducted an OOD experiment. The results in Appendix 5 reveal a crucial trade-off.

---

> ### Author Response · Authors · 2025-12-04
> **Rebuttal Response 2**
>
> W1:
>
> Our experiments focus on a sub-frontier deployment regime (1.7B–14B), which commonly appears in latency- and cost-constrained production settings (edge/on-premise servers, enterprise inference clusters), where serving multiple 70B–100B models simultaneously is often infeasible. In such environments, routing typically happens among mid-sized, specialized models, not exclusively frontier LLMs.
>
> While our experiments use models up to 14B, the framework itself extends naturally to larger pools. Scaling to 70B/100B models would require only retraining the multi-label correctness predictor and re-calibrating CRC, but no changes to the algorithmic design. We will clarify this scope and note that evaluating CR$^2$ on frontier-scale pools is a valuable next step rather than a limitation of the method.
>
> W2:
>
> In our setting, **the questions that smaller models answer correctly are not fully subsumed by the larger model**, as the pool is heterogeneous in capability rather than size-monotonic. Although the 14B model is the largest, several smaller models outperform it on subsets, meaning it does _not_ dominate the pool and cannot serve as the true upper bound.
>
> Crucially, effective routing leverages model complementarity, allowing it to surpass the accuracy of the standalone 14B model. This potential is validated by the Oracle upper bound, which strictly exceeds the performance of any single model in the pool.
>
> W3:
>
> We thank the reviewer for pointing this out and take the concern seriously. We agree that the original phrasing _“consistently outperforms state-of-the-art baselines”_ was stronger than appropriate given the benchmark-level results, and we have revised it to a more precise statement reflecting the data.
>
> While CR² does not outperform all baselines on every individual benchmark, we observe that on **more than half of the benchmark-level Pareto curves, as well as on the aggregated cost–accuracy frontier across all tasks, CR² achieves state-of-the-art trade-offs.** We will add these Pareto comparisons in the revised version, along with per-benchmark breakdowns, so that readers can verify where performance gains occur and under what cost regimes.
>
> We hope this clarifies that the issue was one of wording rather than intention, and that the updated claims and additional result visualizations resolve the concern.
>
> W4:
>
> Thank you for highlighting this — the original figure can indeed be misinterpreted. Figure 3b shows an aggregated cost–accuracy frontier across all benchmarks, but this was not clearly stated. In the revised version, we now provide:
>
> 1. per-benchmark Pareto frontiers for CR² and all baselines in Figure 4,
>
> 2. aggregated frontiers with baseline curves overlaid, rather than single averaged points,
>
> 3. an explicit caption noting that the main panel is an aggregate view.
>
> These updates make the trade-offs across benchmarks directly visible and allow readers to compare CR² against baselines on both per-task and global Pareto fronts.
>
> W5:
>
> As requested, we evaluated correctness prediction using a simple linear head on top of the embeddings. The results show that a linear classifier trained on SCL embeddings performs similarly to one trained on the original embedding space, indicating that SCL does not primarily improve _linear_ separability of correctness labels.
>
> However, the t-SNE visualization in Fig. 1(c) reveals a clearer and more structured embedding geometry after contrastive learning. This suggests that SCL influences the organization of the representation space rather than increasing linear discriminability along a single projection. The ablation results (Appendix 3) further support this interpretation: removing SCL leads to a performance drop in routing despite similar linear probe accuracy, demonstrating that its benefit lies in geometric structuring rather than direct linear separation.
>
> W6:
>
> We are currently in the final stages of refactoring and documenting the codebase to ensure it is user-friendly and easy to run. We guarantee to make the code publicly available via GitHub upon paper acceptance. In the meantime, we hope our detailed responses to the specific technical questions have sufficiently clarified the implementation details.

---

### Meta-Review · Area_Chair_gg6z · 2026-01-05

**Summary:**

The paper proposes a method for model routing based on conformal risk control (CRC). The basic idea is to employ supervised contrastive learning to learn query embeddings, upon which a multi-label CRC classifier is built.

The original reviews raised a number of concerns:
- **Clarity**. Two reviewers noted that the presentation of the core methodology has scope for improvement.
- **Restriction to binary quality scores**. Two reviewers noted that given increased interest in generative tasks, it is important to consider tasks such as text generation.
- **Experimental setup**. The lack of frontier models, apparent ability of small LMs to out-perform LMs, and concerns around stability of results were noted as potential issues.
-  **Reporting of experimental results**. The claimed gains of the method are not in line with the results presented.
- **Novelty of CRC**. It was noted that there is prior work on conformal risk control for model routing, which had not been cited and discussed.

**Reviewer Concerns:**

- **Clarity**. The authors made some clarifications on specific points that reviewers noted as being unclear (e.g., role of $t_1$). However, the deeper issue in the overall presentation is challenging for a response alone to resolve.
  - *Partially addressed*. In my reading, I agree with the reviewers that this is an issue that requires a number of edits. Further to the reviewers' comments, we make a few notes.
    - Some of the references are missing a space following the preceding text, e.g., "GPT(OpenAI, 2025b)".
    - The distinction of existing routing methods into learning-based and similarity-based makes sense. The latter could however be seen as employing a non-parametric learner. Note e.g., that a $k$-NN method does require some labelled examples.
    - Speculative decoding is a related but conceptually quite distinct framework to model routing (e.g., the large model must be called for every token, unlike query- or token-routing).
    - Figure 3 appears three pages before the results are discussed. Please reconsider the arrangement of results.
- **Restriction to binary quality scores**. The authors suggested that the method can be easily extended to handle soft scores in $[0, 1]$, although no results were presented for this setting.
    - *Partially addressed*. It would have been compelling to see results verifying that the framework does indeed provide gains in one such setting. However, it does seem plausible that the framework can be extended to the case of generic quality metrics.
- **Experimental setup**. The authors suggested that the focus on smaller models was intentional, targeting on-device regimes. They also suggested that the ability of small models to outperform larger ones on data subsets was expected.
  - *Partially addressed*. The arguments provided seemed reasonable. Further to the reviewers' comments, we make a few notes.
    - It is surprising that EmbedLLM was claimed as a SoTA _routing_ method: that paper presented routing as one application of a general recipe for building _LLM_ embeddings. Works such as RouteLLM are more typically understood to be SoTA for model routing.
    - It is also surprising that RouteDC did not feature more discussion, given the apparent similarities to the supervised CL embedding proposed in this paper. It is not conceptually clear whether certain representation learning approaches will be superior for the CRC framework.
- **Reporting of experimental results**. The authors agreed that the original presentation was misleading, and presented a re-wording. They also claimed the main conclusion holds, namely, that the proposed method does outperform baselines on average.
   - *Partially addressed*. The proposed re-wording seems reasonable. The response refers to updated per-task Pareto curves, which would need to be assessed to fully verify the authors' claims.
- **Novelty of CRC**. The authors described points of distinction over the prior work, such as the training on an explicit classifier over embeddings rather than reliance on confidence scores.
  - *Mostly addressed*. The distinction requires further discussion in a revised version. We also note that the paper's discussion of related work on model routing is fairly narrow; e.g., see Somerstep et al., _CARROT: A Cost Aware Rate Optimal Router_; Jitkrittum et al., _Universal Model Routing for Efficient LLM Inference_; Chuang et al., _Learning to route LLMs with confidence tokens_; and references therein.

**Reviewer Scores:**

- **QVqS**: as some of the concerns were addressed, we believe it likely that the score of 0 would have been raised. But given there may still be some outstanding concerns per above, we tend to think it would be at most 4.
- **bn9g**: given the primary concern was around clarity, and given the brevity of the response, we tend to think the score would remain as-is at 4.
- **RSU5**: as the reviewers' concerns were generally addressed and not as severe as the others, we could imagine the score being raised to slightly above the accept bar to 6.
- **gDhA**: given the primary concern was around clarity, and given the brevity of the response, we tend to think the score would remain as-is at 4.

---

### Decision · Program_Chairs · 2026-01-26

Reject